# Kidney Stone Disease: Epigenetic Dysregulation in Homocystinuria and Mitochondrial Sulfur Trans-Sulfuration Ablation Driven by COVID-19 Pathophysiology

**DOI:** 10.3390/biom15081163

**Published:** 2025-08-14

**Authors:** Anmol Babbarwal, Mahavir Singh, Utpal Sen, Mahima Tyagi, Suresh C. Tyagi

**Affiliations:** 1Department of Epidemiology and Population Health, School of Public Health and Information Sciences (SPHIS), University of Louisville, Louisville, KY 40202, USA; 2Department of Physiology, School of Medicine, University of Louisville, Louisville, KY 40202, USA; utpal.sen@louisville.edu (U.S.); suresh.tyagi@louisville.edu (S.C.T.); 3Lady Hardinge Medical College, Shaheed Bhagat Singh Marg, Connaught Place, New Delhi 110001, India; mahimatyagi2106@gmail.com

**Keywords:** spike protein, acute kidney injury, homocysteine, trans-sulfuration pathway, epigenetics, kidney stone disease

## Abstract

The coronavirus disease 2019 (COVID-19) pandemic, caused by severe acute respiratory syndrome coronavirus 2 (SARS-CoV-2), has brought to light unexpected complications beyond respiratory illness, including effects on kidney function and a potential link to kidney stone disease (KSD). This review proposes a novel framework connecting COVID-19-induced epigenetic reprogramming to disruptions in mitochondrial sulfur metabolism and the pathogenesis of kidney stones. We examine how SARS-CoV-2 interferes with host methylation processes, leading to elevated homocysteine (Hcy) levels and impairment of the trans-sulfuration pathway mechanisms particularly relevant in metabolic disorders such as homocystinuria. These epigenetic and metabolic alterations may promote specific kidney stone subtypes through disrupted sulfur and oxalate handling. Additionally, we explore the role of COVID-19-associated gut dysbiosis in increasing oxalate production and driving calcium oxalate stone formation. Together, these pathways may accelerate the transition from acute kidney injury (AKI) to chronic KSD, linking viral methylation interference, sulfur amino acid imbalance, mitochondrial dysfunction, and microbiota changes. Unlike earlier reviews that address these mechanisms separately, this work offers an integrated hypothesis to explain post-viral renal lithogenesis and highlights the potential of targeting sulfur metabolism and redox pathways as therapeutic strategies for KSD triggered or aggravated by viral infections such as COVID-19.

## 1. Introduction

The outbreak of SARS-CoV-2, the etiological agent of COVID-19, has profoundly altered global health dynamics, leaving behind a persistent trail of long-term complications, collectively referred to as post-COVID sequelae [1,2]. While the acute manifestations of COVID-19 have been widely studied, it is becoming increasingly evident that the virus’s impact on organ systems persists well beyond initial recovery, contributing to diverse chronic conditions [3,4,5,6,7]. Recent clinical reports and retrospective cohort studies have raised concerns about a potential rise in renal complications, including an unexpected uptick in kidney stone presentations among post-COVID patients [8,9,10]. Although large-scale population-based studies are still emerging, preliminary data suggest an increased incidence of urolithiasis in individuals recovering from SARS-CoV-2 infection, particularly those who experienced renal involvement during the acute phase [8]. This underscores the urgent need to investigate post-viral renal sequelae beyond the immediate window of AKI [5,11,12,13]. It is noteworthy that dietary supplementation during the COVID-19 pandemic in the form of vitamin C (ascorbic acid) gained attention as a possible treatment or preventive measure, and it is known that vitamin C can lead to elevated levels of urinary oxalate that increase the risk of hyperoxaluria and the formation of oxalate kidney stones [14,15,16].

It is important to mention that the trans-sulfuration pathway is a metabolic route that converts the non-proteinogenic amino acid Hcy into cysteine through a series of enzymatic steps, primarily involving the enzymes cystathionine β-synthase (CBS) and cystathionine γ-lyase (CSE) [17]. This pathway plays a crucial role in regulating Hcy levels and producing cysteine, which is essential for synthesizing glutathione, a major antioxidant in the body [18,19,20,21]. In the kidneys, the proper function of the trans-sulfuration pathway supports redox balance, vascular tone, and cellular detoxification processes [22]. Disruption of this pathway leads to elevated Hcy, oxidative stress, and impaired nitric oxide (NO) signaling, all of which have been linked to endothelial dysfunction and renal injury [23,24]. In the context of KSD, such metabolic disturbances can create an environment that favors crystal formation and stone development [15,16,25].

Current therapeutic interventions focus primarily on immune modulation and RNA-based antiviral strategies, yet these approaches offer limited protection against long-term metabolic, vascular, and renal complications. Among the most concerning consequences is the link between COVID-19 and the progression of chronic kidney disease (CKD), including the development of KSD [26,27,28]. Despite growing clinical observations, the molecular mechanisms driving this transformation from AKI to chronic stone-forming nephropathy remain highly elusive. Our review proposes a novel, mitochondria-centered mechanism connecting COVID-19 infection to the onset of KSD [10]. We postulate that disruptions in mitochondrial sulfur metabolism, specifically within the trans-sulfuration pathway, play a key role in kidney stone pathogenesis, particularly in the context of viral infection and immune dysregulation. This pathway, essential for Hcy clearance and cysteine production, becomes compromised during COVID-19 due to viral interference with epigenetic and bioenergetic cellular machinery [28,29]. Elevated homocysteine (Hcy) levels, a condition referred to as hyperhomocysteinemia (HHcy), may intensify systemic inflammation and directly facilitate stone formation by encouraging the crystallization of sulfur-containing amino acids within the kidneys.

Moreover, COVID-19-induced metabolic reprogramming characterized by enzyme succinylation, mitochondrial dysfunction, and microbial dysbiosis creates a powerful biochemical milieu that is favorable to nephrolithiasis [30,31]. For instance, gut microbiota such as *Escherichia coli* degrades tricarboxylic acid (TCA) intermediates into oxalate, thereby facilitating calcium oxalate stone formation [32,33]. Simultaneously, viral proteins and therapeutic antibodies have the potential to enhance immune activation and endothelial damage, thus setting the stage for micro-stone formation [8,10,15,16].

Mechanistically, viral entry into host cells is mediated by the spike (S) protein of SARS-CoV-2, which binds to angiotensin-converting enzyme 2 (ACE2) receptor [34,35] (Figure 1). This interaction is not merely a gateway for infection; it also initiates a cascade of cellular stress responses that can compromise mitochondrial function, redox balance, and renal epithelial integrity [36,37]. Hence, understanding the downstream effects of viral-host interactions is critical for designing mechanism-based therapeutic interventions that go beyond the acute phase and target the root causes of long-term renal complications, including kidney stone formation [10]. By integrating insights from virology, renal physiology, epigenetics, and mitochondrial biology, our review aims to chart a conceptual framework linking post-COVID metabolic disruptions to kidney stone disease. In doing so, we hope to pave the way for novel diagnostic markers and targeted therapies for patients at risk of long-term nephrological complications after COVID-19 infection. Unlike previous reviews that have focused either on viral-induced kidney injury or metabolic changes in isolation, this manuscript integrates epigenetic hijacking, sulfur metabolism impairment, and renal immune–metabolic crosstalk as a unified pathway from COVID-19 to kidney stone formation [5,10,15,16].

## 2. Epigenetics of COVID-19 Infection and KSD

### 2.1. Epigenetic Methylation Dysregulation and SARS-CoV-2

The epigenetic landscape plays a pivotal role in modulating host responses to SARS-CoV-2, particularly within renal tissues [38,39,40]. A central regulatory node involves DNA and histone methylation, which directly shapes gene expression, immune surveillance, and metabolic processing. Methionine adenosyl transferase (MAT) synthesizes S-adenosylmethionine (SAM), the principal methyl donor for DNA, RNA, protein, and histone methylation [41,42]. Viral infection disrupts this tightly regulated balance since SARS-CoV-2 encodes its own methyltransferase (MT) that hijacks the host methylation machinery, skewing normal methylation rhythms and leading to elevated production of Hcy as a toxic byproduct [43,44]. Impaired SAM availability alters DNA methyltransferase (DNMT) activity, and dysregulation of TET enzymes can collectively modify chromatin accessibility and inflammatory gene expression. Importantly, methylation changes such as increased H3K4me3 and RNA N6-methyladenosine (m^6^A) marks have been implicated in sustaining inflammatory loops and oxidative stress, both of which can promote tubular damage, fibrosis, and a stone-promoting milieu (Figure 2) [15,16,42].

### 2.2. Hcy and Sulfur Metabolism: Implications for KSD

While cysteine promotes calcium oxalate (CaOx) crystal formation [45], HHcy, which is common in severe or post-COVID patients, may exacerbate renal lithogenesis through distinct sulfur-mediated mechanisms [42]. HHcy arises from disrupted methylation cycles and impaired mitochondrial trans-sulfuration, leading to intracellular Hcy accumulation [43,44]. Although the precise role of Hcy in crystallization remains poorly defined, its chemical reactivity may foster calcium-phosphate and sulfur-based crystal formation, particularly in inflamed or hypoxic renal microenvironments. Hcy may also induce renal epithelial injury via oxidative stress, NO depletion, and endothelial dysfunction, each contributing to crystal adhesion and fibrotic remodeling [46]. These hypotheses underscore the need for targeted experimental studies evaluating Hcy’s contribution to crystal nucleation and retention in COVID-19-affected kidneys.

### 2.3. Immune-Mediated Endothelial Injury and Fibrosis

Following SARS-CoV-2 entry through ACE2 and TMPRSS2, ACE2 downregulation alters the renin–angiotensin system (RAS), increasing Ang II levels while depleting protective Ang-(1–7) peptides [47,48,49]. This imbalance contributes to systemic hypertension, endothelial stress, and AKI, which may transition to CKD and predispose to KSD [50,51]. COVID-19-induced coagulopathy, despite normal platelet counts, reflects microvascular injury rather than classical thrombosis [41,52]. A notable factor is a reduction in the activity of a Disintegrin and Metalloproteinase with a Thrombospondin Type 1 motif, member 13 (ADAMTS13), a metalloproteinase that regulates von Willebrand factor and vascular homeostasis [53,54,55]. Its suppression promotes endothelial dysfunction, vascular stiffness, and ECM fragmentation, laying the groundwork for glomerular injury, proteinuria, and tubulointerstitial fibrosis. Our prior studies in hACE2 mice intranasally challenged with SARS-CoV-2 SP revealed upregulation of pro-inflammatory M1 macrophages and activation of proteolytic enzymes such as TMPRSS2, neutrophil gelatinase-associated lipocalin (NGAL), ADAMTS13, and matrix metalloproteinases (MMPs) (Figure 2) [52,56,57]. These proteases participate in ECM remodeling and fibrosis, while Neopterin (NPT), a downstream product of M1 macrophage activity, depletes tetrahydrobiopterin (BH4), reducing endothelial nitric oxide synthase (eNOS) activity and vasodilation capacity. Subsequent peroxynitrite (ONOO^−^) formation drives further oxidative damage, metalloproteinase activation, and endothelial barrier dysfunction. Collectively, these pathways disrupt glomerular integrity and promote urinary stasis and lithogenesis [15,16].

### 2.4. Dysfunction and miRNA-Mediated Reprogramming

Mitochondrial dysfunction is a hallmark of COVID-19 and is linked to renal injury and fibrotic progression. Elevated levels of NGAL and FGF23, both implicated in tubular injury and vascular calcification, are observed in post-COVID syndromes [58]. These changes are potentiated by neutrophil extracellular traps (NETs), which amplify renal damage via glycocalyx shedding and capillary rarefaction [59,60]. Single-cell transcriptomics confirm activation of ADAM17, a sheddase associated with focal segmental glomerulosclerosis. Male patients often show higher expression of TMPRSS2 and ADAM17, correlating with worse outcomes [61,62]. Hence, inhibitors targeting these proteases may hold therapeutic value. Furthermore, solute carriers such as SLC22A17, which complex with NGAL, regulate iron, cadmium, and zinc transport and are implicated in nephrotoxicity, osmotic stress adaptation, and renal inflammation [11,63]. Activation of MMP8 and release of microparticles during COVID-19 further damage the perivascular niche and glomerular architecture [64].

### 2.5. From Biomarkers to Therapeutics: Neopterin and iNOS

We propose a unified mechanistic model in which the SP primes ACE2/TMPRSS2 and activates M1 macrophages, driving iNOS upregulation, BH4 depletion, and peroxynitrite formation. These changes lead to endothelial dysfunction and trigger cascades involving NETs, NGAL, FGF23, MMPs, and ADAMTS proteases, the key mediators of glomerular leakage, vascular stiffness, and stone-promoting fibrosis [7]. The iNOS inhibition or genetic knockout (iNOSKO) significantly reduces these outcomes in experimental models, reinforcing the translational potential of anti-inflammatory therapeutics. In this context, neopterin evolves from a mere biomarker to a mechanistic driver of renal damage, suggesting its potential as a therapeutic target (Table 1).

**Table 1 biomolecules-15-01163-t001:** Key Biomarkers Implicated in COVID-19-Associated KSD. The candidate biomarkers relevant to the pathophysiological link between SARS-CoV-2 infection and kidney stone formation are shown. Listed biomarkers include indicators of oxidative stress, immune activation, mitochondrial dysfunction, and impaired sulfur metabolism. Each marker is annotated with its biological source (e.g., urine or plasma), functional relevance, and potential utility in diagnosing or monitoring post-COVID-19 renal complications.

Biomarker	Source	Relevance
Homocysteine (Hcy)	Plasma/Urine	Elevated in HHcy; pro-oxidant; stone-promoting
NGAL	Urine/Plasma	Early kidney injury marker; linked to COVID-19
BH_4_ (Tetrahydrobiopterin)	Blood/Urine	Reflects redox imbalance and NOS uncoupling
Neopterin	Urine/Serum	Marker of macrophage activation, immune stress
NETs	Plasma	Linked to both COVID-19 and renal inflammation

### 2.6. Epigenetics, Viral Persistence, and Long-Term Risk of KSD

Beyond acute infection, SP persistence and chronic cytokine release sustain low-grade inflammation, mirroring features of autoimmune and fibrotic disease [65,66,67,68,69]. Epigenetic dysregulation continues post-COVID, including histone methylation changes, miRNA activation (e.g., miR-21), and mitochondrial miRNA-2392 mediated suppression of oxidative phosphorylation [70,71]. These modifications reprogram host metabolism toward glycolysis and hypoxia, the conditions that impair renal repair and may favor crystal formation and retention [15,16].

## 3. Kidney Stones

Kidney stones, also known as renal calculi, are hardened mineral deposits that develop in the renal tubules and collecting system when urine becomes supersaturated with specific solutes. They are generally categorized into two main types according to their appearance on radiographic imaging: radiopaque and radiolucent. Of the radiopaque variety, calcium-based stones, especially those composed of calcium oxalate and calcium phosphate, are the most common. In contrast, uric acid-containing calculi are the most common radiolucent stones [72]. The urinary pH plays a pivotal role in stone formation. Alkaline urine (pH ≥ 7.0) is associated with the precipitation of calcium phosphate and struvite stones, while acidic urine (pH ≤ 5.3) promotes the crystallization of uric acid and cystine stones [72,73]. Importantly, the biochemical and molecular pathogenesis of different stone types is distinct, involving diverse pathways of mineral metabolism, microbial dysbiosis, and epithelial dysfunction. A particularly compelling mechanism is the microbial degradation of the TCA cycle, which has implications in kidney stone pathophysiology. For instance, a pathogenic strain of *Escherichia coli* has been shown to degrade citrate (a TCA intermediate with anti-lithogenic properties) into dicarboxylic acids such as oxalate [73]. Oxalate is a key constituent of calcium oxalate stones, and such microbial-driven conversion from TCA to DCA intermediates provides a plausible mechanistic link between gut dysbiosis and lithogenesis. However, whether gut dysbiosis post-COVID-19 infection plays a causative role in kidney stone formation remains to be systematically studied. Although these findings are compelling, direct clinical evidence linking *E. coli*-mediated citrate degradation to increased urinary oxalate in post-COVID patients is still limited. Most existing data are derived from in vitro microbial metabolism studies or animal models. Thus, the hypothesis that *E. coli* may contribute to post-infectious lithogenesis via citrate catabolism into oxalate requires further validation. This may be especially relevant in post-COVID-19 individuals, where microbial dysbiosis and altered host–microbe metabolic interactions have been increasingly reported. Prospective studies examining urinary citrate and oxalate levels in relation to specific bacterial colonization patterns will be necessary to substantiate this mechanistic pathway.

Of additional concern is the observed increase in HHcy in patients suffering from severe COVID-19 and long COVID-19 symptoms, including pneumonia and renal complications [42,74,75]. Hcy is a sulfur-containing amino acid that, when elevated, exerts pro-inflammatory and pro-oxidative effects that may disrupt renal epithelial integrity, facilitate tubular injury, and increase crystalluria. Within the mitochondria, the trans-sulfuration pathway converts Hcy to hydrogen sulfide (H_2_S), a gasotransmitter with protective vasodilatory and antioxidant effects. The 3-mercaptopyruvate sulfurtransferase (3MST), a mitochondrial enzyme, catalyzes this reaction, and its activity is vital for maintaining endothelial function and mitochondrial redox balance in renal tissues. The impairment of mitochondrial function has emerged as a central feature in post-COVID-19 sequelae, including kidney disease and potentially kidney stone formation. Mitochondrial dysfunction, especially due to disrupted bioenergetics and altered post-translational modifications of key enzymes, is implicated in altered cellular homeostasis in the renal microenvironment. A notable modification is lysine succinylation, which occurs robustly on TCA cycle enzymes during COVID-19 infection [30]. This modification impairs enzymatic function and disrupts energy production and redox signaling. The mitochondrial desuccinylase SIRT5, which plays a key role in reversing lysine succinylation, is emerging as a critical target for restoring mitochondrial efficiency in the post-viral state [30]. Given the high energy demand and oxidative stress burden in renal epithelial cells, disrupted mitochondrial bioenergetics could create a milieu conducive to stone formation through cellular injury, impaired ion transport, and altered solute handling. Another potentially therapeutic strategy involves pyruvate supplementation, which enhances TCA cycle flux, mitigates oxidative stress, and suppresses viral replication [76]. In the context of COVID-19-induced mitochondrial dysfunction, pyruvate may also restore epithelial barrier integrity and reduce the risk of stone formation by normalizing energy metabolism in the renal tubular epithelium. In this context, KSD can no longer be viewed solely as a local renal pathology but rather as a systemic condition involving epigenetic reprogramming, microbial interactions, and mitochondrial bioenergetics, especially following COVID-19. It is therefore imperative to consider a multi-dimensional therapeutic approach, including restoring mitochondrial function, targeting HHcy through epigenetic and nutritional strategies, and exploring the role of microbiome modulation in the prevention of KSD post-COVID-19.

## 4. Underlying Mechanisms Linking COVID-19 Infection to Kidney Stone Formation

The emergence of KSD following COVID-19 infection appears to involve a multifactorial pathophysiology, encompassing mitochondrial dysfunction, epigenetic alterations, inflammatory cascades, vascular and tubular injury, and disruptions in metabolic homeostasis, particularly within the sulfur amino acid pathway. Central to these processes is mitochondrial sulfur metabolism, which is regulated by enzymes such as 3MST. This enzyme integrates the trans-sulfuration pathway with the TCA cycle, epigenetic regulation, and redox balance [77,78].

### 4.1. Mitochondrial Sulfur Metabolism and Hcy Accumulation

The 3MST is involved in Hcy detoxification and H_2_S generation, which is a gasotransmitter with anti-inflammatory, antioxidant, and vasodilatory properties in renal tissues [79]. In the context of COVID-19, mitochondrial stress and inflammation can impair 3MST activity, leading to elevated Hcy levels and diminished H_2_S synthesis [80]. This state of HHcy is associated with increased oxidative stress, endothelial dysfunction, and thrombotic risk, all of which can contribute to renal microvascular injury and tubulointerstitial fibrosis [81,82]. Additionally, Hcy may crystallize in acidic urine, forming Hcy stones; however, this process requires further investigation in human studies. While Hcy accumulation is associated with renal injury, direct crystallization of Hcy in vivo has not yet been confirmed and warrants further investigation.

### 4.2. Trans-Sulfuration Pathway Disruption and Epigenetic Consequences

The trans-sulfuration pathway comprising CBS, CSE, and 3MST is critical for converting methionine to cysteine and subsequently to H_2_S. SARS-CoV-2 infection disrupts this pathway through mechanisms such as increased methylation demand from viral replication, which depletes SAM, alters methylation capacity, and affects gene regulation. Furthermore, dysfunction of sirtuins (e.g., SIRT5 and SIRT3) interferes with histone succinylation and acetylation, affecting mitochondrial enzyme activity and gene expression linked to renal tubular integrity [83]. These epigenetic shifts may promote fibrotic, inflammatory, and pro-thrombotic gene expression, potentially driving the progression from AKI to CKD [84].

### 4.3. COVID-19-Induced Mitochondrial Dysfunction

SARS-CoV-2 infection has been linked to mitochondrial alterations, including reduced oxidative phosphorylation, altered mitochondrial DNA expression (e.g., downregulation via miR-2392), increased reactive oxygen species (ROS), and mitochondrial membrane depolarization [36]. Reduced NAD^+^ levels also compromise sirtuin-dependent metabolic regulation. These changes impair TCA cycle activity, promote aerobic glycolysis, and support a pro-fibrotic and inflammatory renal microenvironment, especially in proximal tubular epithelial cells, which are central to solute regulation [85]. Such dysfunction may promote urinary supersaturation of calcium, oxalate, uric acid, and phosphate, favoring stone formation [14,15,16,86].

### 4.4. Oxidative and Nitrosative Stress

Excess ROS and peroxynitrite (ONOO^−^), generated in part through inducible nitric oxide synthase (iNOS), contribute to tetrahydrobiopterin (BH_4_) depletion. This results in the uncoupling of endothelial nitric oxide synthase (eNOS) and further exacerbates endothelial injury and renal ischemia. ROS and ONOO^−^ also activate MMPs, ADAMTS13, and NGAL, collectively leading to glycocalyx degradation, glomerular permeability, and renal fibrosis. MMP-driven remodeling may also alter renal architecture and facilitate crystal nidus formation [15,16].

### 4.5. Immune Activation, Macrophages, and NETosis

COVID-19 is characterized by heightened immune responses, including a cytokine storm and M1 macrophage polarization, which promotes NPT production and BH_4_ depletion. Neutrophils in this context release NETs, which entrap urinary crystals and cellular debris, initiating inflammation and promoting stone matrix development [87,88]. The interplay between activated macrophages, NGAL, and ADAMTS in the kidney’s microvasculature contributes to COVID-19-associated coagulopathy (CAC) and may favor calcification and stone nucleation, particularly in distal tubules and collecting ducts [52,88].

### 4.6. Tubular Transport Dysfunction and Osmotic Stress

SARS-CoV-2 infection disrupts solute carriers and ion transporters, such as SLC22A17, which remains in complex with NGAL and is involved in the transport of metal ions (Fe, Cd, and Zn) and tubular endocytosis [89,90]. Impaired transporter function can lead to osmotic imbalance, tubular cell injury, and urinary concentration of lithogenic solutes [88].

### 4.7. Role of Gut Microbiome and Uremic Toxins

Alterations in the gut microbiota following COVID-19 may elevate systemic levels of uremic toxins and oxalate. The loss of oxalate-degrading bacteria like *Oxalobacter formigenes* may increase the risk of calcium oxalate stone formation [91,92]. Although the precise impact of SARS-CoV-2 on gut microbiome composition and its role in enteric hyperoxaluria remains under investigation, emerging evidence suggests a potential link [93,94].

### 4.8. Systemic Hypoxia and Dehydration

Patients with COVID-19 commonly experience dehydration, hypoxemia, and immobility, especially during hospitalization [95,96]. These conditions concentrate urine, reduce urinary citrate excretion, and impair renal perfusion, which are well-established risk factors for uric acid and other types of stones, particularly in acidic environments [45,88]. In summary, SARS-CoV-2 infection initiates a cascade of interrelated molecular and cellular events that may contribute to a pro-lithogenic renal milieu. Central to this process is the impairment of mitochondrial sulfur metabolism, particularly 3MST-mediated conversion of Hcy to H_2_S. Disruption of this pathway results in HHcy, oxidative stress, epigenetic dysregulation, and metabolic alterations. These changes collectively induce glomerular and tubular injury, immune activation, and urinary solute supersaturation, ultimately facilitating renal crystal formation. We believe that a deeper understanding of these mechanisms may offer novel therapeutic strategies, including iNOS inhibition to mitigate nitrosative stress, pyruvate supplementation to support mitochondrial function, BH4 restoration to improve endothelial integrity, use of epigenetic modulators, and microbiome-based interventions [97]. Considering the global prevalence of COVID-19 and its possible lasting effects on kidney function, systematically exploring its contribution to kidney stone formation remains a critical focus for both nephrology and public health research.

## 5. Potential Limitations and Future Directions

The COVID-19 pandemic has revealed a wide range of both immediate and long-term health complications linked to SARS-CoV-2 infection. While its respiratory effects are well understood, there is a pressing need to better define the virus’s systemic impacts, particularly on kidney health and its possible role in KSD. Most existing studies focus on viral entry mechanisms and immune responses, but far less attention has been given to non-infectious outcomes such as S protein-driven intracellular signaling, gut microbiome disturbances, and changes along the renal axis. These processes may be key contributors to the development of KSD after COVID-19 and therefore warrant thorough investigation. Our hypothesis for pathogenic remodeling by the S protein of SARS-CoV-2, particularly its interaction with angiotensin-converting enzyme 2 (ACE2) and transmembrane serine protease 2 (TMPRSS2), describes how the S protein helps initiate the viral entry into the host cells. However, this binding cascade may also trigger a pro-inflammatory and pro-oxidative signaling network independent of direct viral replication. Studies suggest that S protein alone is sufficient to elicit endothelial dysfunction, oxidative stress, and pro-fibrotic responses [30,52]. Thus, the S protein may act as a viral toxin, activating the inducible nitric oxide synthase (iNOS) resulting in the overproduction of nitric oxide (NO) and peroxynitrite (ONOO^−^), NPT production through M1 macrophage polarization, NET formation, NGAL release from injured tubular cells, and activation of MMPs and ADAMTS family members, including ADAM17, which is implicated in both shedding of ACE2 and pro-inflammatory cytokine activation [52,98,99]. This molecular cascade promotes disruption of the glomerular and tubular glycocalyx, epithelial leakage, collagen deposition, glomerulosclerosis, and tubulointerstitial fibrosis, all potential precursors to a potential renal lithogenesis process [100]. Thus, these mechanisms offer a biologically plausible explanation for the emergence of KSD following COVID-19 and hence merit targeted experimental validation.

Emerging evidence highlights the essential role of the gut–kidney axis in renal health. SARS-CoV-2 infection significantly alters the composition and function of the intestinal microbiota, often leading to gut dysbiosis. This microbial imbalance can lead to increased oxalate absorption due to depletion of *Oxalobacter formigenes*, elevated production of uremic toxins such as p-cresol sulfate and indoxyl sulfate, and compromised short-chain fatty acid (SCFA) biosynthesis that affects systemic inflammation and renal epithelial integrity [72,73]. These alterations influence renal stone formation both directly by increasing urinary oxalate, and indirectly, by promoting systemic inflammation and oxidative stress. Furthermore, dysbiosis-driven endotoxemia may exacerbate the inflammatory state in COVID-19, amplifying renal injury via the Toll-like receptor (TLR)–NFκB pathway, thereby intensifying cytokine storm effects and contributing to stone nucleation in a sensitized renal environment [100]. We opine that targeting the M1-iNOS pathway to mitigate renal injury might prove to be a novel and testable hypothesis emerging from this work because a targeted blockade of M1 macrophage-induced iNOS activity as a therapeutic intervention to prevent COVID-19-induced renal injury and lithogenesis. Further, inhibiting iNOS could restore BH4 availability and reduce oxidative/nitrosative stress, suppress NPT production and NET formation, reduce NGAL expression and ADAMTS activation, preserve glycocalyx architecture and reduce endothelial permeability, and prevent maladaptive fibrosis and crystal deposition [52,100,101,102]. The link between NET formation and renal lithogenesis in post-viral states remains speculative but represents an important direction for future research. Experimental models using selective iNOS inhibitors or BH4 supplementation could elucidate these mechanisms. Additionally, RNAi knockdown of TMPRSS2 and ADAM17, particularly in sex-differentiated models, may also provide insight into gender disparities observed in COVID-19 severity and renal complications [30,52]. Indeed, androgen-mediated upregulation of TMPRSS2 in males may underlie the increased severity and mortality reported during the pandemic, making TMPRSS2 inhibition a viable sex-specific strategy. To address the hypotheses mentioned above and bridge knowledge gaps, we would like to propose the following research directions.

### 5.1. Animal Models

Use S protein or pseudo viruses to simulate non-replicative renal injury. Assess iNOS, NPT, NGAL, and NETs expression in kidneys. Evaluate the efficacy of iNOS inhibitors and pyruvate supplementation [76]. Also, one can use transgenic mice with renal cell-specific 3MST knockout to examine sulfur metabolism’s role in post-viral injury.

### 5.2. Epigenetic Analyses

Perform ChIP-seq and mass spectrometry to profile histone succinylation and acetylation changes in kidneys post-infection. Investigate how SIRT5 modulation affects fibrosis, inflammation, and mitochondrial metabolism [30].

### 5.3. Clinical Biomarker Studies

Measure plasma and urinary levels of Hcy, NGAL, NPT, NETs, BH4, and ADAMTS13 in post-COVID-19 patients with and without kidney stones. Correlate these biomarkers with renal function, imaging studies, and stone composition (Table 1).

### 5.4. Gut Microbiome Sequencing

Conduct longitudinal microbiota analyses pre- and post-COVID-19 in KSD-prone individuals. Investigate restoration strategies such as probiotics, fecal microbiota transplantation, or oxalate-degrading bacterial therapies.

### 5.5. Therapeutic Trials

Pilot trials using pyruvate, BH4 analogs, or SIRT activators in long COVID patients with early signs of renal dysfunction. Examine the effect of androgen receptor antagonists or TMPRSS2 inhibitors on male patients with recurrent stones or long COVID nephropathy.

### 5.6. Sex-Differentiated Renal Analysis

Evaluate gene and protein expression levels of ACE2, TMPRSS2, ADAM17, and related pathways in male versus female renal tissues. Study the protective role of estrogens or selective estrogen receptor modulators (SERMs) in modulating post-COVID-19 renal injury.

## 6. Conclusions

While this conceptual framework is biologically plausible, several limitations must be acknowledged. Causality between COVID-19 and KSD remains speculative and primarily based on associative data, but emerging epidemiological reports underscore a notable uptick in nephrological complications, including kidney stones, among post-COVID populations. For example, recent studies suggest an increased incidence of renal colic, AKI, and recurrent nephrolithiasis in long COVID cohorts, particularly in patients with predisposing metabolic syndromes or immune dysregulation [9,103,104]. Large-scale health system data analysis and insurance claims databases have also begun to report higher than expected rates of new onset renal calculi following COVID-19 infection, especially among middle-aged and elderly individuals [105]. However, these findings remain under-validated and warrant prospective studies to confirm them [106]. The longitudinal course of lithogenesis post-COVID-19 is poorly understood due to the lack of imaging follow-ups or urine biochemistry in post-viral cohorts. Animal models may not fully recapitulate human renal tubular physiology or immune responses to SARS-CoV-2 proteins. Biomarker assays (e.g., NETs and BH_4_) lack standardization across clinical laboratories, hence limiting reproducibility (Table 1). The spike protein’s off-target toxicity in non-infective settings, while mechanistically compelling, remains under-characterized. Despite these limitations, the proposed hypotheses and research avenues offer a transformative approach to understanding the renal sequelae of COVID-19. Given the chronic and recurrent nature of KSD and the expanding global population of long COVID patients estimated at over 65 million individuals worldwide as of recent WHO estimates [107,108], there is an urgent need to recognize and study non-traditional post-viral complications such as nephrolithiasis. Early identification of metabolic, immunological, and epigenetic drivers may help prevent irreversible kidney damage and improve long-term outcomes.

COVID-19 has expanded our perception of viral infections beyond simple contagion. Its capacity to alter host metabolic networks, induce persistent immune disturbances, and reshape epigenetic regulation positions it as a disorder involving complex systemic dysfunction. This review sheds light on an emerging connection among mitochondrial sulfur metabolism, homocysteine (Hcy) biology, epigenetic changes, and kidney stone formation, advocating for more extensive research into less obvious consequences like kidney stone disease (KSD). We suggest that nephrology in the coming decade must adopt this broader perspective, combining knowledge from virology, immunology, metabolic science, and epigenetics to better understand and address the prolonged impacts of COVID-19 in at-risk populations. Moreover, prioritizing large cohort studies, incorporating KSD data into long COVID registries, and focusing on specific biomarker assessments in post-COVID patients will be essential steps toward translating these mechanistic findings into clinical practice (see Table 1).

## Figures and Tables

**Figure 1 biomolecules-15-01163-f001:**
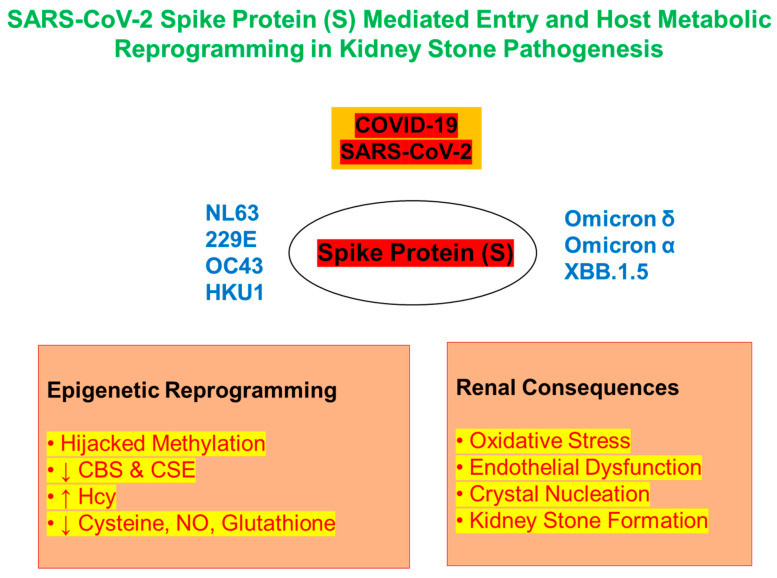
Viral entry of SARS-CoV-2 and human coronaviruses via spike protein (SP)-mediated mechanisms. The SP of SARS-CoV-2, including emerging variants such as Omicron-α, Omicron-δ, and XBB.1.5, and other human coronaviruses (NL63, 229E, OC43, and HKU1) enables viral entry into host cells by binding surface receptors and initiating fusion. After entry, SARS-CoV-2 hijacks the host’s methylation machinery, leading to epigenetic reprogramming and altered metabolic pathways. One key consequence is disruption of the mitochondrial sulfur trans-sulfuration pathway, which normally converts Hcy to cysteine via CBS and CSE enzymes. This disruption leads to Hcy accumulation, depletion of cysteine and glutathione, and impaired NO production, contributing to oxidative stress and renal endothelial dysfunction. These changes create a pro-stone environment in the kidney, linking COVID-19 pathogenesis to the development of KSD [10,15,16].

**Figure 2 biomolecules-15-01163-f002:**
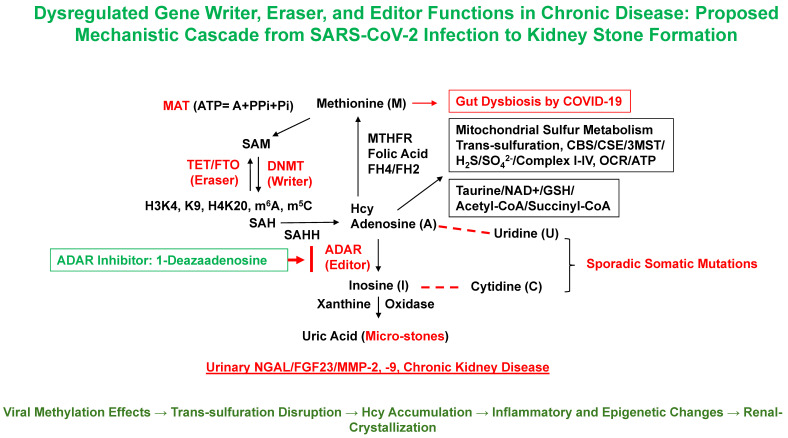
Gut dysbiosis-induced epigenetic and metabolic reprogramming linking COVID-19 to renal pathology. The schematic illustrates the hypothesized cascade by which SARS-CoV-2-associated gut dysbiosis contributes to KSD and chronic kidney injury. SARS-CoV-2 infection disrupts the gut microbiota, impairing folate-mediated one-carbon metabolism and mitochondrial sulfur trans-sulfuration pathways. These changes lead to altered levels of key metabolites such as Hcy, SAM, and glutathione (GSH), which in turn influence the activity of epigenetic regulators: gene “writers” (e.g., DNMTs), “erasers” (e.g., TET demethylases and FTO), and “editors” (e.g., ADAR enzymes). Dysregulated epigenetic modifications such as aberrant histone methylation (H3K4me3, H3K9me3, and H4K20me3) and RNA methylation (m^6^A and m^5^C) affect nuclear–mitochondrial crosstalk and downregulate mitochondrial oxidative phosphorylation components (Complexes I–IV), reducing oxygen consumption rate (OCR) and ATP production. This bioenergetic decline, coupled with increased oxidative stress and inflammation, contributes to renal tubular remodeling, crystal retention, and formation of uric acid micro-stones. The figure also highlights potential diagnostic biomarkers associated with this pathway such as NGAL, fibroblast growth factor 23 (FGF23), and MMP-2/-9, which are elaborated in Table 1. Together, this integrative model links gut dysbiosis, epigenetic reprogramming, and metabolic derangement to post-COVID kidney stone disease [15,16].

## Data Availability

This manuscript is a review article, so none of the data were generated or analyzed. All data discussed were sourced from previously published studies, which have been appropriately cited in the text.

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
