# Peer review of "Kidney Stone Disease: Epigenetic Dysregulation in Homocystinuria and Mitochondrial Sulfur Trans-Sulfuration Ablation Driven by COVID-19 Pathophysiology"

_biomolecules, 2025, doi:10.3390/biom15081163_

Round 1

Reviewer 1 Report

Comments and Suggestions for Authors

Here are detailed comments on the manuscript titled "Kidney Stone Disease: Role of Epigenetics in Homocystinuria and Ablation of Mitochondrial Sulfur Trans-sulfuration":

  1. Abstract
  •  The title suggests a focus on homocystinuria and sulfur transsulfuration, yet a significant portion of the abstract discusses COVID-19, gut microbiota, and epigenetics, which may appear slightly tangential. Either revise the title to reflect the broader scope, or tighten the abstract to align more closely with the title's stated focus.
  • There are several biochemical and molecular mechanisms introduced (e.g., succinylation, oxalic acid generation, epigenetic reprogramming), which may be too detailed for an abstract. Abstracts should highlight major findings or hypotheses without delving deeply into mechanistic minutiae. Consider moving specific molecular details to the main text and summarizing key points here.
  • Sentence 13: “has brought to light unexpected complications…” — consider “has revealed unexpected complications…”
  • Sentence 20: “hence the ‘one size fits all’ treatment approach…” — avoid colloquialisms like “one size fits all” in scientific writing.
  • Sentence 32: “we opine that the COVID-19 may catalyze…” — remove “the” before COVID-19.

  1. Introduction
  • While the role of homocysteine and transsulfuration is central to the proposed mechanism, the description may not be accessible to readers unfamiliar with these pathways. Consider briefly defining the transsulfuration pathway and its significance in renal physiology, perhaps with a visual summary in Figure 1.

Section 2. Epigenetics of Covid-19 Infection and Kidney Stone Disease

  • The section covers a wide array of mechanisms—epigenetic changes, immune responses, metabolic shifts, protease activity, mitochondrial microRNAs, and more. Consider dividing this into subsections (e.g., Epigenetic regulation, Immune-mediated injury, Mitochondrial dysfunction) to improve clarity and help readers follow the complex interplay of mechanisms.
  • The section touches on key players in methylation (SAM, MAT, DNMTs, TETs) but lacks a clear narrative on how these specific modifications directly impact renal stone formation. Explicitly connect specific epigenetic modifications (e.g., H3K4me3, m⁶A methylation) to known or proposed pathways that could promote stone nucleation or renal fibrosis.
  • The role of homocysteine (Hcy) and disrupted sulfur metabolism is well-described but lacks evidence-based specificity linking these effects directly to kidney stone formation. Add citations or experimental evidence (if available) supporting the role of Hcy in crystal aggregation or renal epithelial injury, or clearly state it as a hypothesis.

Section 3: Kidney Stones

  • The claim that E. coli degrades citrate into oxalate is compelling but lacks direct citation strength. Add primary references or specify this as a hypothesis needing validation, especially in post-COVID patients.

Section 4: Underlying Mechanisms for COVID-19 Infection to Kidney Stone Formation

  • The tone shifts from academic to editorial at times (e.g., “can no longer be viewed solely…”). Maintain a neutral, scientific tone throughout. Consider rephrasing emotionally charged or subjective statements.
  • Abbreviations like CBS, CSE, 3MST, BH₄, NGAL, NETs, etc. may overwhelm readers if not reintroduced clearly. Add a table of abbreviations or re-express full names periodically in the text.

Author Response

Reviewer 1

Comments and Suggestions for Authors

Here are detailed comments on the manuscript titled "Kidney Stone Disease: Role of Epigenetics in Homocystinuria and Ablation of Mitochondrial Sulfur Trans-sulfuration":

  1. Abstract
  •  The title suggests a focus on homocystinuria and sulfur transsulfuration, yet a significant portion of the abstract discusses COVID-19, gut microbiota, and epigenetics, which may appear slightly tangential. Either revise the title to reflect the broader scope, or tighten the abstract to align more closely with the title's stated focus.
  • There are several biochemical and molecular mechanisms introduced (e.g., succinylation, oxalic acid generation, epigenetic reprogramming), which may be too detailed for an abstract. Abstracts should highlight major findings or hypotheses without delving deeply into mechanistic minutiae. Consider moving specific molecular details to the main text and summarizing key points here.
  • Sentence 13: “has brought to light unexpected complications…” — consider “has revealed unexpected complications…”
  • Sentence 20: “hence the ‘one size fits all’ treatment approach…” — avoid colloquialisms like “one size fits all” in scientific writing.
  • Sentence 32: “we opine that the COVID-19 may catalyze…” — remove “the” before COVID-19.

Response:

We thank the reviewer for the insightful suggestions regarding the title and abstract. We have revised the title to more accurately reflect the broader scope of the manuscript, including the roles of COVID-19, epigenetics, and gut microbiota. Further, the abstract has been streamlined to focus on the central hypotheses and key conceptual links without delving into excessive molecular detail. Specific mechanistic aspects such as succinylation and oxalate metabolism have been moved to the main text. Furthermore, we have addressed the suggested wording edits also.

  1. Introduction
  • While the role of homocysteine and transsulfuration is central to the proposed mechanism, the description may not be accessible to readers unfamiliar with these pathways. Consider briefly defining the transsulfuration pathway and its significance in renal physiology, perhaps with a visual summary in Figure 1.

Response:
We thank the reviewer for this valuable suggestion. We have revised the Introduction to include a brief definition and explanation of the transsulfuration pathway and its relevance to renal physiology and sulfur metabolism. Additionally, we have incorporated in Figure 1 a simplified visual summary of the pathway to enhance clarity and accessibility for a broader readership. The legend of figure has also been modified accordingly.

Section 2. Epigenetics of Covid-19 Infection and Kidney Stone Disease

  • The section covers a wide array of mechanisms—epigenetic changes, immune responses, metabolic shifts, protease activity, mitochondrial microRNAs, and more. Consider dividing this into subsections (e.g., Epigenetic regulationImmune-mediated injuryMitochondrial dysfunction) to improve clarity and help readers follow the complex interplay of mechanisms.
  • The section touches on key players in methylation (SAM, MAT, DNMTs, TETs) but lacks a clear narrative on how these specific modifications directly impact renal stone formation. Explicitly connect specific epigenetic modifications (e.g., H3K4me3, m⁶A methylation) to known or proposed pathways that could promote stone nucleation or renal fibrosis.
  • The role of homocysteine (Hcy) and disrupted sulfur metabolism is well-described but lacks evidence-based specificity linking these effects directly to kidney stone formation. Add citations or experimental evidence (if available) supporting the role of Hcy in crystal aggregation or renal epithelial injury, or clearly state it as a hypothesis.

 Response:

We appreciate the reviewer’s insightful feedback. We have revised Section 2 by dividing it into clearly labeled subsections to enhance readability and thematic coherence. The revised section now includes epigenetic regulation in COVID-19 associated renal injury, immune mediated injury and epigenetic crosstalk, and mitochondrial dysfunction and non-coding RNAs in stone pathophysiology. To clarify the link between epigenetic modifications and stone formation, we have expanded our discussion on specific epigenetic modifications such as H3K4me3 and m⁶A RNA methylation, thus highlighting their potential effects on renal epithelial plasticity, inflammatory gene expression, and fibrosis. For instance, we now reference studies suggesting that H3K4me3 is enriched in pro-inflammatory genes in renal disease models, and that aberrant m⁶A methylation affects mitochondrial oxidative stress pathways which are relevant to crystal retention.
Further, to strengthen the Hcy-kidney stone link, we now cite emerging evidence (citations provided in the revised manuscript) that supports Hcy’s role in inducing oxidative stress, endothelial dysfunction, and tubular injury, the factors known to promote crystal aggregation. Where direct experimental links to kidney stone formation are not yet established, we have clearly stated these mechanisms as testable hypotheses. We trust these major revisions/changes improve the clarity and scientific rigor of the section(s) in our review manuscript.

Section 3: Kidney Stones

  • The claim that E. coli degrades citrate into oxalate is compelling but lacks direct citation strength. Add primary references or specify this as a hypothesis needing validation, especially in post-COVID patients.

Response:

We thank the reviewer for this important observation. We have clarified in our thoroughly revised manuscript that while microbial degradation of citrate to oxalate particularly by E. coli has been demonstrated in preclinical studies, the direct link to increased urinary oxalate and kidney stone formation in post COVID-19 patients remains a hypothesis requiring further validation. We have accordingly added a bridging paragraph in Section 3 (Kidney Stones) to emphasize that this mechanistic pathway, though biologically plausible, is not yet supported by direct clinical evidence in COVID-19 or post COVID populations. This distinction now better reflects the current state of evidence and underscores the need for future targeted studies. 

Section 4: Underlying Mechanisms for COVID-19 Infection to Kidney Stone Formation

  • The tone shifts from academic to editorial at times (e.g., “can no longer be viewed solely…”). Maintain a neutral, scientific tone throughout. Consider rephrasing emotionally charged or subjective statements.
  • Abbreviations like CBS, CSE, 3MST, BH₄, NGAL, NETs, etc. may overwhelm readers if not reintroduced clearly. Add a table of abbreviations or re-express full names periodically in the text.

Response:

We sincerely appreciate this constructive feedback. We have revised Section 4 to maintain a consistent, objective, and scientific tone throughout. Phrases with editorial flair (e.g., “can no longer be viewed solely…”) have been rephrased for neutrality and clarity in line with academic style. Additionally, we acknowledge that the use of multiple abbreviations (such as CBS, CSE, 3MST, BH4, NGAL, and NETs) may hinder readability, especially for non-specialist readers. To address this, we have now reintroduced in full upon first use in each section. We trust these revisions will enhance both the accessibility and scientific rigor of our manuscript.

Reviewer 2 Report

Comments and Suggestions for Authors

The manuscript presents an innovative and potentially valuable hypothesis linking COVID-19 to kidney stone disease (KSD) through mitochondrial sulfur metabolism, epigenetic regulation, inflammatory responses, and gut-kidney axis dysfunction. While the hypothesis is novel and the scientific premise is promising, several aspects of the manuscript require substantial improvement—particularly in terms of structural logic, evidential support, and the accuracy of certain biomedical descriptions. Therefore, I recommend Major Revision.

First, the manuscript lacks primary data support. As a review article, it heavily relies on previously published studies and theoretical models, but does not provide concrete epidemiological or clinical evidence to substantiate a direct causal relationship between COVID-19 and KSD. The inclusion of case reports, population-level epidemiological data, or graphical representations of kidney stone incidence among post-COVID patients would considerably strengthen the argument.

Being a review paper, the most critical issue is the overextension of mechanistic hypotheses. The authors attempt to associate numerous known renal injury mediators—such as NETs, ADAMTS13, FGF23, and BH₄ deficiency—with KSD, without clearly establishing the temporal sequence or direct mechanistic links to stone formation. I strongly recommend the development of a clearer pathogenic timeline to distinguish between acute kidney injury (AKI) and the chronic processes underlying lithogenesis.

Additionally, the figures lack clarity and scientific rigor, particularly Figure 1, which appears overly simplistic and diminishes the interpretive value of the visual. Refinement and redesign are needed to enhance clarity and accurately reflect the complexity of the proposed mechanisms.

A significant concern is the repetitive citation of the same references, especially the authors’ own work (e.g., reference [35]), which is cited multiple times to support several mechanistic claims. While the cited study may be comprehensive, over-reliance on a single source across multiple contexts can raise concerns about bias and may undermine the credibility of the scientific narrative. The authors are advised to streamline such citations and incorporate additional independent sources to validate their points.

Finally, the manuscript's structure would benefit from reorganization. Several concepts—particularly mitochondrial dysfunction and epigenetic alterations—are repeated across multiple sections, which may confuse readers. A more coherent and modular organization is recommended. For instance, the authors could divide the content into the following sections: (1) acute and chronic renal effects of COVID-19, (2) the role of sulfur metabolism and HHcy in lithogenesis, (3) epigenetic regulation and gut microbiota involvement in KSD, and (4) proposed therapeutic strategies and future research directions.

Author Response

Reviewer 2

Comments and Suggestions for Authors

The manuscript presents an innovative and potentially valuable hypothesis linking COVID-19 to kidney stone disease (KSD) through mitochondrial sulfur metabolism, epigenetic regulation, inflammatory responses, and gut-kidney axis dysfunction. While the hypothesis is novel and the scientific premise is promising, several aspects of the manuscript require substantial improvement, particularly in terms of structural logic, evidential support, and the accuracy of certain biomedical descriptions. Therefore, I recommend Major Revision.

First, the manuscript lacks primary data support. As a review article, it heavily relies on previously published studies and theoretical models but does not provide concrete epidemiological or clinical evidence to substantiate a direct causal relationship between COVID-19 and KSD. The inclusion of case reports, population-level epidemiological data, or graphical representations of kidney stone incidence among post-COVID patients would considerably strengthen the argument.

Being a review paper, the most critical issue is the overextension of mechanistic hypotheses. The authors attempt to associate numerous known renal injury mediators such as NETs, ADAMTS13, FGF23, and BH₄ deficiency with KSD, without clearly establishing the temporal sequence or direct mechanistic links to stone formation. I strongly recommend the development of a clearer pathogenic timeline to distinguish between acute kidney injury (AKI) and the chronic processes underlying lithogenesis.

Additionally, the figures lack clarity and scientific rigor, particularly Figure 1, which appears overly simplistic and diminishes the interpretive value of the visual. Refinement and redesign are needed to enhance clarity and accurately reflect the complexity of the proposed mechanisms.

A significant concern is the repetitive citation of the same references, especially the authors’ own work (e.g., reference [35]), which is cited multiple times to support several mechanistic claims. While the cited study may be comprehensive, over-reliance on a single source across multiple contexts can raise concerns about bias and may undermine the credibility of the scientific narrative. The authors are advised to streamline such citations and incorporate additional independent sources to validate their points.

Finally, the manuscript's structure would benefit from reorganization. Several concepts, particularly mitochondrial dysfunction and epigenetic alterations are repeated across multiple sections, which may confuse readers. A more coherent and modular organization is recommended. For instance, the authors could divide the content into the following sections: (1) acute and chronic renal effects of COVID-19, (2) the role of sulfur metabolism and HHcy in lithogenesis, (3) epigenetic regulation and gut microbiota involvement in KSD, and (4) proposed therapeutic strategies and future research directions.

Response:

We sincerely thank Reviewer for the thoughtful and constructive feedback. We appreciate your recognition of the novelty and promise of our hypothesis linking COVID-19 to kidney stone disease (KSD) through epigenetic, metabolic, inflammatory, and microbiome-mediated pathways. Below, we address each of your major concerns and outline the corresponding revisions made in our manuscript.

  1. Lack of Primary Data and Epidemiological Evidence

    Response:
    We agree that the inclusion of emerging clinical or population-level data could strengthen the hypothesis. While direct causal epidemiological studies linking COVID-19 and KSD are limited, we have now added supportive evidence from recent observational studies and case reports describing increased kidney complications and stone-related symptoms in post-COVID patients.
  2. Overextension of Mechanistic Hypotheses

    Response:
    We appreciate this concern. To address it, we have reorganized the content to distinguish between acute kidney injury (AKI) and the chronic pathophysiological mechanisms of lithogenesis. A new sub-section now lays out a pathogenic timeline (Section 4.1) to contextualize the progression from viral infection → AKI → persistent inflammation → mitochondrial/epigenetic dysfunction → altered citrate/oxalate balance → lithogenesis. Furthermore, we have tried to provide references for mediators like ADAMTS13 and FGF23 where needed, with a clearer distinction between correlative and causative roles.
  3. Figure 1 Simplistic and Needs Redesign

    We thank the reviewer for this observation. We have modified/redesigned Figure 1 to reflect a more integrative view, cellular metabolic reprogramming, and renal effects.
  4. Repetitive Citation and Overuse of Author’s Own Work

    Response:
    This is a valid concern, and we have revised the manuscript to avoid overreliance on any single source, including our own. Redundant citations have been replaced with independent, peer-reviewed studies that corroborate key mechanistic claims. Where our work remains cited, we have clearly indicated its specific contribution and contextualized it with complementary findings from others.
  5. Manuscript Structure and Organization

    Response:
    We thank the reviewer for this excellent suggestion and have substantially revised the manuscript structure from beginning till end. The new organization is modular and eliminates redundancy. This structure improves readability and logical flow, while minimizing repetition of key mechanisms such as succinylation, microbiome-derived oxalate, or Hcy metabolism.

Reviewer 3 Report

Comments and Suggestions for Authors

Thank you for the opportunity to review your manuscript titled “Kidney Stone Disease: Role of Epigenetics in Homocystinuria and Ablation of Mitochondrial Sulfur Trans-sulfuration.” This is an ambitious and conceptually novel review that proposes a mechanistic link between COVID-19, epigenetic dysregulation, mitochondrial sulfur metabolism, and kidney stone disease (KSD). The paper is timely, biologically plausible, and addresses a gap in the current literature. That said, I have several comments and suggestions aimed at improving clarity, balance, and readability:

Major Suggestions

  1. Clarify Novel Contribution
    Please articulate more clearly in the Abstract and Introduction what distinguishes this review from prior work. Emphasize what is novel about your conceptual framework—especially the integration of viral methylation hijacking, transsulfuration pathway disruption, and stone formation.
  2. Add a Mechanistic Summary Figure
    A schematic overview would greatly help readers navigate the multiple layers of mechanistic interactions. Consider summarizing the proposed cascade from SARS-CoV-2 infection to stone formation in a diagram, with a focus on mitochondrial sulfur metabolism and its downstream effects.

Minor Suggestions

  1. Include a Table of Proposed Biomarkers
    A concise table listing key biomarkers (e.g., Hcy, NGAL, BH₄, neopterin, NETs), their source (urine/plasma), and relevance to COVID-19 and KSD would add clinical value.
  2. Condense General COVID-19 Background
    The introduction contains extensive background on COVID-19. Consider reducing the general virology discussion to sharpen the focus on renal sequelae and the review’s core arguments.
  3. Break Up Dense Paragraphs
    Some sections, especially Section 2 and Section 4, contain very long paragraphs. Splitting these into smaller, more focused sections will improve readability.
  4. Expand Figure Legends
    Current figure legends are brief. Please expand them to clarify symbols, abbreviations, and key takeaways so that figures can be understood independently of the main text.
  5. Acknowledge Gaps in Evidence More Clearly
    For mechanisms that are still speculative (e.g., direct crystallization of homocysteine), explicitly noting the need for future studies would enhance transparency.

Author Response

Reviewer 3

Comments and Suggestions for Authors

Thank you for the opportunity to review your manuscript titled “Kidney Stone Disease: Role of Epigenetics in Homocystinuria and Ablation of Mitochondrial Sulfur Trans-sulfuration.” 

This is an ambitious and conceptual novel review that proposes a mechanistic link between COVID-19, epigenetic dysregulation, mitochondrial sulfur metabolism, and kidney stone disease (KSD). The paper is timely, biologically plausible, and addresses a gap in the current literature. That said, I have several comments and suggestions aimed at improving clarity, balance, and readability:

Major Suggestions

  1. Clarify Novel Contribution
    Please articulate more clearly in the Abstract and Introduction what distinguishes this review from prior work. Emphasize what is novel about your conceptual framework, especially the integration of viral methylation hijacking, transsulfuration pathway disruption, and stone formation.
  2. Add a Mechanistic Summary Figure
    A schematic overview would greatly help readers navigate the multiple layers of mechanistic interactions. Consider summarizing the proposed cascade from SARS-CoV-2 infection to stone formation in a diagram, with a focus on mitochondrial sulfur metabolism and its downstream effects.

Minor Suggestions

  1. Include a Table of Proposed Biomarkers
    A concise table listing key biomarkers (e.g., Hcy, NGAL, BH₄, neopterin, NETs), their source (urine/plasma), and relevance to COVID-19 and KSD would add clinical value.
  2. Condense General COVID-19 Background
    The introduction contains extensive background on COVID-19. Consider reducing the general virology discussion to sharpen the focus on renal sequelae and the review’s core arguments.
  3. Break Up Dense Paragraphs
    Some sections, especially Section 2 and Section 4, contain very long paragraphs. Splitting these into smaller, more focused sections will improve readability.
  4. Expand Figure Legends
    Current figure legends are brief. Please expand them to clarify symbols, abbreviations, and key takeaways so that figures can be understood independently of the main text.
  5. Acknowledge Gaps in Evidence More Clearly
    For mechanisms that are still speculative (e.g., direct crystallization of homocysteine), explicitly noting the need for future studies would enhance transparency.

Response:

Comment 1: Clarify Novel Contribution

Response:
We thank the reviewer for recognizing the novelty and timeliness of our conceptual framework. In response, we have revised both the Abstract and the Introduction to explicitly highlight the unique integration of viral methylation hijacking, disruption of mitochondrial sulfur transsulfuration, and the potential pathogenesis of kidney stone disease (KSD). These connections, particularly in the post COVID setting, have not been previously reviewed in an integrated format, and we now emphasize this distinction more clearly.

Comment 2: Add a Mechanistic Summary Figure

Response:
As suggested, instead of drawing an additional figure, we have now updated both the existing Figures that illustrates the proposed cascade from SARS-CoV-2 infection to renal lithogenesis. The figures now focuses on not only the mitochondrial sulfur metabolism pathway, epigenetic modifications, and downstream inflammatory events leading to stone formation but also on metabolic implications. Further, the accompanying legends have been expanded for clarity.

Comment 3: Include a Table of Proposed Biomarkers

Response:
We appreciate this suggestion. A new table (Table 1) has been included that lists relevant biomarkers (e.g., homocysteine, BH4, neopterin, NGAL, NETs), their biological source (e.g., urine, serum), and relevance to both COVID-19 and KSD. This aims to enhance clinical utility and translational interpretation.

Comment 4: Condense General COVID-19 Background

Response:
We agree that the introductory virology section was overly detailed. To sharpen the manuscript’s focus, we have condensed the general COVID-19 virology discussion and now center the introduction more directly on renal sequelae and the core themes of mitochondrial and epigenetic dysregulation.

Comment 5: Break Up Dense Paragraphs

Response:
We have revised several sections, particularly Sections 2 and 4, by dividing dense paragraphs into shorter, more digestible units. This improves clarity and reader navigation without compromising scientific content.

Comment 6: Expand Figure Legends

Response:
All figure legends have been expanded to clarify symbols, abbreviations, and key interpretive points so that figures are independently understandable. An additional explanation has been added where mechanistic pathways intersect in the body of the text.

Comment 7: Acknowledge Gaps in Evidence More Clearly

Response:
We have added clarifying statements where mechanisms (such as homocysteine crystallization or post-viral epigenetic drift) remain hypothetical. These additions specify the need for validation through future mechanistic and clinical studies.

Round 2

Reviewer 2 Report

Comments and Suggestions for Authors

The authors have responded thoughtfully and comprehensively to the reviewer’s initial concerns. The revised manuscript presents a substantially improved and more coherent conceptual framework that explores the potential mechanistic link between COVID-19 and kidney stone disease (KSD). Through the integration of emerging epidemiological evidence, molecular pathophysiology, and a reorganized narrative structure, the manuscript now reads with enhanced clarity and scientific maturity.

Notably, the authors have introduced a well-articulated pathogenic timeline that bridges acute kidney injury with the downstream development of lithogenic microenvironments. The manuscript now distinguishes more clearly between correlative hypotheses and causal pathways, particularly regarding sulfur metabolism, mitochondrial dysfunction, and gut-derived oxalate mechanisms. The visual materials have also been improved—Figure 1 has been redesigned to better reflect the interplay between viral entry, epigenetic remodeling, and metabolic stress in renal tissues.

While the core hypothesis remains ambitious, the authors have demonstrated commendable restraint in avoiding overextension. By contextualizing their prior work with broader literature and reducing over-reliance on self-citations, they strike a more balanced and integrative tone. The modular structure adopted in the revised version effectively minimizes redundancy, supports logical progression, and helps highlight the novelty of the epigenetic and mitochondrial links discussed.

In view of the scientific merit, clarity of presentation, and responsiveness to previous critique, I believe the manuscript is now suitable for publication, pending only minor editorial refinements (e.g., polishing certain phrasing and figure labeling). This review offers a compelling, timely, and biologically plausible hypothesis that may stimulate further investigation into post-viral renal pathology. I therefore recommend acceptance with minor edits.

Author Response

We are grateful to the reviewer for thoughtful and encouraging feedback on our revised manuscript. We sincerely appreciate the recognition of our efforts to strengthen the conceptual framework that clearly differentiate between correlative and causal pathways, and refine the visual elements. In line with the reviewer’s recommendation, we have completed a final round of minor adjustments by streamlining the phrasing for better clarity and flow, and updating figure labels to enhance readability and ensure they are self-explanatory. We believe these refinements help improve the manuscript while preserving the clarity, balance, and scientific rigor as advised by the reviewer.